# Wear Resistance and Titanium Adhesion of Cathodic Arc Deposited Multi-Component Coatings for Carbide End Mills at the Trochoidal Milling of Titanium Alloy

**Marina A. Volosova \*, Sergey V. Fyodorov, Stepan Opleshin and Mikhail Mosyanov**

Department of High-Efficiency Processing Technologies, Moscow State University of Technology "STANKIN", Vadkovsky per. 1, 127055 Moscow, Russia; sv.fedorov@icloud.com (S.V.F.); stepanoplesnin@gmail.com (S.O.); mmosyanov@yandex.ru (M.M.)
\* Correspondence: m.volosova@stankin.ru

**Abstract:** The work was devoted to the study of the effectiveness of the application of multi-component coatings, TiN–Al/TiN, TiN–AlTiN/SiN, and CrTiN–AlTiN–AlTiCrN/SiN, obtained by cathodic arc deposition to increase the wear resistance of 6WH10F carbide end mills in trochoidal milling of titanium alloy. The surface morphology of the tool with coatings was studied using scanning electron microscopy, and surface roughness texture was estimated. Microhardness and elastic modulus of the coated carbide tool surface layer were determined by nanoindentation. The process of sticking titanium to the working surface of the tool and quantitative evaluation of end mill wear with multi-component coatings at the trochoidal strategy of milling titanium alloy was studied. The CrTiN–AlTiN–AlTiCrN/SiN coating showed the maximum value of the plasticity index at the level of 0.12. The maximum effect of reducing the wear rate was achieved when using a tool with a CrTiN –AlTiN–AlTiCrN/SiN coating when the operating time to failure of end mills was increased by 4.6 times compared to samples without coating, by 1.4 times compared with TiN–Al/TiN coating and 1.15 times compared with TiN–AlTiN/SiN coating.

**Keywords:** multi-component coatings; cathodic arc deposition; end mills; trochoidal milling; titanium alloys; wear resistance; machining efficiency; surface morphology; microhardness; titanium sticking; nanolayer; nanocomposite

## 1. Introduction

Titanium alloys have become indispensable structural materials in the aerospace and aviation industries due to the optimal ratio of strength properties, weight characteristics, excellent corrosion, and heat resistance over a wide temperature range. Aircraft frame elements, disks, vanes of engines and compressors are just some examples of widely used parts made of titanium-based alloys. Despite these undeniable advantages, titanium alloys have certain disadvantages, primarily poor machinability that encourages plenty of researchers to use various techniques to improve titanium alloy machining [1–3]. First, the high specific strength of titanium alloys leads to an increase in temperature in the cutting zone. Secondly, titanium is chemically active, which causes the activation of adhesive processes, welding, and sticking of chips to the tool. The tool wears out intensively with this combination of power and thermal loads, and often cutting-edge chipping is observed during milling [4–7].

Various approaches can be applied to increase the efficiency of milling of titanium alloys that can be associated with optimization of the machining strategy, such as improvement of design and

geometric parameters [3–11] as well as increased wear resistance of the surface layer of the tool due to the application of special coatings [1,2,12–15].

Currently, the use of a trochoidal milling strategy to remove a large volume of chips during processing of titanium alloy is considered as one of the most advanced machining methods that have not been fully studied [16–18]. The leading manufacturers of metal cutting tools such as Sandvik Coromant, Seco Tools, and others remark the vast potential of trochoidal milling for processing aerospace materials and offer end mills of an innovative design that demonstrate outstanding results conditions of real production [19,20].

The tool moves along a circular path in a trochoidal milling strategy programmed by an intelligent multi-axis machine control system with each circumferential path moving forward in the direction of general motion (Figure 1). One of the key features of trochoidal milling is that only a small area of the end mill is involved in the engagement at the same time, and the feed rate is always constant. It allows using higher cutting speeds in combination with the significant volumetric performance due to the reduced and redistributed even cutting load and, consequently, the whole length of the mill working part can be involved in processing. It helps as well to reduce wear in comparison with the standard strategies of milling. Besides, trochoidal milling allows using a carbide end mill with a diameter less than the width of the groove to be machined, which is extremely important to ensure the possibility of unhindered removal of titanium chips from the cutting zone. Despite its potential, the trochoidal milling creates specific problems for technologists [18,19,21]. The carbide end mill makes a complex movement, which should be specially programmed in the CNC system. In addition, the elements of a metal-cutting machine as a spindle unit, technological equipment, etc. should be sufficiently rigid and have an operating speed so that it can be used for a trochoidal milling strategy. The carbide end mill should also have a particular configuration, improved physical and mechanical properties to ensure milling at high cutting speeds, feed size, and cutting depth [22–25]. However, the release of a large amount of heat during machining of heat-resistant titanium alloys limits the cutting speed.

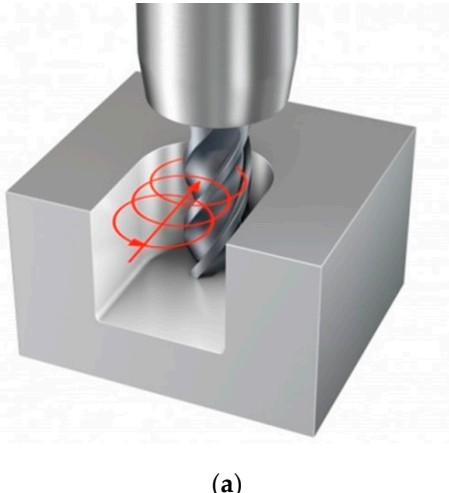
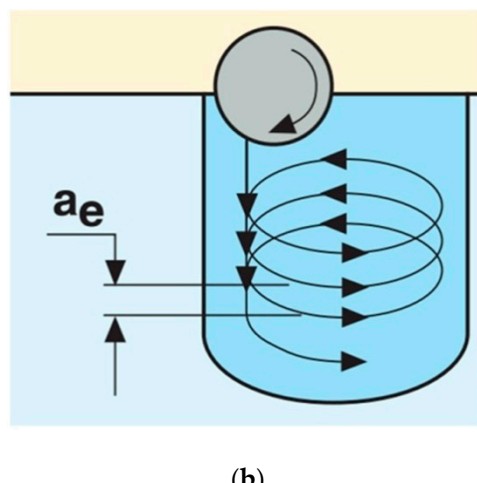

(**a**)                    (**b**)

**Figure 1.** Schematic diagram of the groove processing in the trochoidal milling strategy: (**a**) the isometric view of the end mill trajectory (Sandvik); (**b**) the end mill trajectory in the plane with the radial depth of cut ($a_e$).

Practice shows that the cutting tool is one of the key elements of the "machine tool-fixture-tool-part" technological system. Even the most precise machine equipment used in the trochoidal milling strategy will not show the expected effect with the insufficient wear resistance of the cutting tool, and the quality of the part will not be provided at the required high level. It is necessary to ensure that the cutter will maintain its operability for the required time to provide high-performance machining with the required quality of the part. Thin wear-resistant coatings based on nitrides of refractory metals (TiAlN,

TiZrN, TiCrN, TiNbAlN, etc.) have been a traditional and well-proven tool for increasing tool wear resistance for many years [26,27]. As can be seen, some of the coatings under certain conditions provide better exploitation properties, including wear resistance, than others [28,29], and should be chosen following the machining conditions [30]. At the same time, transition-metal nitride coatings [31–33] and multi-component and nanostructured coatings [33–37] have proven their superiority in the case of hard-to-machine material processing. Practical examples are well known when the wear resistance of a carbide tool increased 6–8 times due to the thin coatings compared to a tool without coatings when machining structural steels [30,36,37] and cast irons [38–40]. However, well-proven coatings do not show the expected effect under the conditions of increased heat and power loads experienced by end mills during the machining of titanium alloys. The specific features of the titanium alloy machining should be taken into account when designing coating technologies. The priority should be given not only to the composition and structure of coatings to be applied but also to the deposition technology [41–44]. Traditional and already well-proven coatings may also not meet expectations in the proposed conditions due to the specifics of the existing loads when implementing a trochoidal milling strategy that can be attended not only to the milling of titanium alloy [17,18,45] but other hard-to-machine materials [46–49].

This study aimed to research the effectiveness of the use of multi-component coatings obtained by cathodic arc deposition to increase the wear resistance of carbide end mills with a trochoidal milling strategy for a titanium alloy.

The scientific novelty of the work is in the comprehensive research of the developed multi-component coatings based on TiN–Al/TiN, TiN–AlTiN/SiN, and CrTiN–AlTiN–AlTiCrN/SiN systems behavior obtained by cathodic arc deposition technology in the experimental conditions which are close to the real tool industry with the purpose to increase the wear resistance of hard alloy end mills during trochoidal milling of titanium alloys that becomes actual regarding wide-spreading and an ever-increasing development of processing technologies of high-strength structural and heat-resistant alloys having their own unique specifics and extraordinary corrosion resistance and are of great interest for modern air- and spacecraft, and other applications.

The tasks of the study included the deposition of the developed coatings on the prepared end tool surface; investigation of the surface morphology of the coated end mill using scanning electron microscopy, estimation of surface roughness; nanoindentation of microhardness and determining elastic modulus of the tool surface layer; research of titanium sticking process to the surface of end mill, quantitative evaluation of end mill wear at the trochoidal milling of titanium alloy.

## 2. Materials and Methods

### 2.1. Research Object and Technology of Coating Deposition

The four-tooth milling cutters (Figure 2) with diameter of 12 mm, length of 140 mm, cutting part of 65 mm, and spiral angle of 40° were used as an object for testing during research. The tungsten–cobalt hard alloy 6WH10F (WC—90%, Co—10%) with hardness of HRA 92.1 (according to Rockwell) and density of 14.50 g/cm$^3$ was used as a tool material.

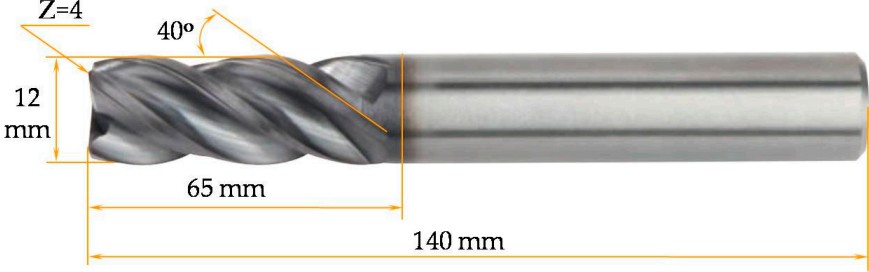

**Figure 2.** Design of the carbide end mill used during experiments.

The technology of cathodic arc deposition of coatings in vacuum was used in this work to improve the wear resistance of end mills during trochoidal milling of a titanium alloy. The high degree of ionization of the vacuum-arc plasma allows using magnetic or electric fields to control the plasma flow, to regulate within a wide range of its energy content, shape, and direction of motion. This allows actively influencing the structural characteristics of coatings, providing the ability to synthesize coatings with specified physical, mechanical, and operational properties. The vast technological capabilities of cathodic arc deposition, combined with excellent performance, are attractive for the formation of nanostructured coatings [50–53]. However, the high-energy intensity of the cathode surface, due to the physics of the appearance and maintenance of the arc discharge at low pressures (~1.5 Pa and below) in the vacuum chamber leads to the appearance of erosion products in the form of macrodrops, which negatively affects the performance of the coated tool [54–57].

An effective solution to the problem of reducing microdrops in the coating was used to eliminate this negative phenomenon—the original LARCs (lateral rotating cathodes) technology developed by Platit AG (Selzach, Switzerland). This technology does not involve the use of bulky magnetic separators but combines the advantages of rotating cylindrical cathodes with their placement on the periphery of the chamber and provides for the use of so-called "virtual shutters" operating without any mechanical elements. The indicated technical solution significantly reduces the number of microdrops in the coating and their roughness due to the direction of the arc plasma in the initial phase of its development of the arc (the initial phase is characterized by an increased content of microdrops) in the opposite direction (to the wall of the vacuum chamber). At the same time, it is important to emphasize that high deposition rates of coatings, which are characteristic for the cathodic arc deposition method, are maintained. The end mill coatings were deposited using a PLATIT $\pi$311 vacuum-arc technological unit (Platit AG, Selzach, Switzerland). The unit was provided with a module of etching technology (lateral glow discharge) that allows plasma etching with argon and metal ion etching (Ti, Cr).

It should be noted that a uniform distribution of plasma in industrial PVD units could be achieved by installing the products to be processed on carrousel with double planetary rotation. Therefore, a uniform distribution of coating can be ensured, since the plasma density in a vacuum chamber without a rotating carrousel can vary by up to 50%.

## 2.2. Justification for the Choice of Coatings and Features of Coating Deposition

The analytical research showed that there are plenty of works related to the wear resistance of the carbide end mill with monolayer coatings during processing of various materials [17,18,45–49], and very few are devoted to testing multilayer- and nanocomposite-structured coatings, especially with the innovative chemical and structural content when the complexity of the coating plays a key role in its resistance.

The complexity of the coating structure hampers the development of cracks that simply cannot develop through the structure of the coating due to the designed obstacles in the form of multiple layers of coating and nanocrystals in an amorphous matrix of the nanocomposite [36,37]. In these conditions, an arising crack cannot straightly cross the complex coating chipping the significant part of the coating up to the substrate as it can be seen in the case of the monolayer coatings (Figure 3a). In the case of a multi- or nanolayered structure, it stops with the first layer, develops further along the diffusion line between the layers, and splits tiny pieces of each layer (Figure 3b). In the case of nanocomposite multilayered coatings, the crack hampers by the crystalline grains that change the direction of crack development in amorphous matrix (Figure 3c). A more complex nanostructure of the coating makes the development of the crack even more difficult (Figure 3d).

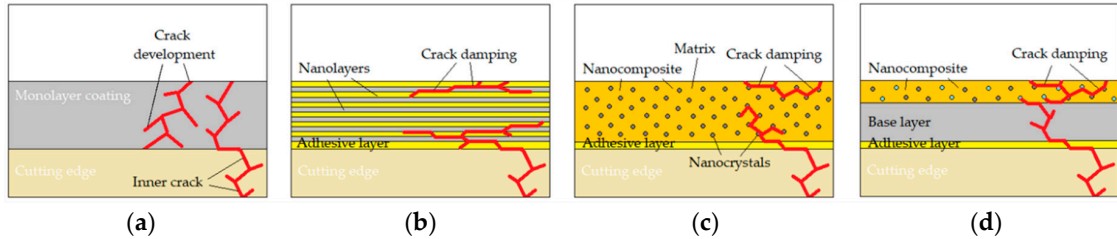

**Figure 3.** The character of crack development in monolayer and multi-component coatings: (**a**) monolayer coating; (**b**) nanolayered coating with adhesive layer; (**c**) nanocomposite coating with adhesive layer; (**d**) multilayered nanocomposite coating.

Three types of multi-component coatings were selected for research based on the proposed analysis and the latest achievements in the field of coatings for tooling purposes [2,30,32–34,36–40,50,58–63]:

(1) The TiN–Al/TiN with a sandwich-type multilayer structure with a TiN adhesive sublayer. Compared to traditional single-layer AlTiN coatings, the TiN–Al/TiN multilayer coating has a higher viscosity and is able to absorb microcracks between the layers. It should be noted that the multi-component coatings usually combine intermediate hard and soft upper and lower layers. Therefore, the crack propagation in multilayer coatings goes along a rather complex and extended path that can be described as complicated. It is related to the theory that a crack is hampered by encountering barriers in its path. Therefore, we can talk about a change in viscosity among the multi-component coating or, in other words, improved viscosity of the multilayer coatings in comparison with monolayer coatings that correspond to the basic concepts of the theory of elasticity [36]. The concept of absorption was used here as a figurative meaning, because the multilayered structure of coating hampers prolongation of the crack. Instead of a direct transversal crack through the monolayer coating towards the substrate surface, a more complex nature can be observed, as it was mentioned above. When a crack traverses the first layer of the multi-component coating, it is hampered by another layer at the place of diffusion of two different layers. It can proceed along the border between the layers and split a single layer of the coating from the rest (wedging spallation). Alternatively, it can traverse the next layer and meet obstacles by another layer of a different material. The more layers in a coating, the more complicated character a crack has, and the more difficult it is to observe the deep cracks in the coatings the shorter they become. Therefore, the mentioned multi-component coating was chosen to protect end mills experiencing increased heat stress during the processing of titanium alloy.

(2) TiN–AlTiN/SiN with a nanocomposite structure and an adhesive TiN sublayer. Mixing of materials (Ti, Al, and Si) does not occur when they are applied. However, two phases are formed—nanocrystalline AlTiN grains embedded in the amorphous SiN matrix. The specified nanocomposite structure significantly improves the physical and mechanical properties of the coating, which shows high wear resistance when cutting difficult-to-machine materials including titanium alloys.

(3) CrTiN–AlTiN–AlTiCrN/SiN with a three-layer nanocomposite structure. The coating is formed in three stages: adhesive sublayer CrTiN; base layer in the form of well-proven AlTiN coatings; the wear-resistant surface layer of high hardness based on AlTiCrN/SiN nanocomposite. Ti, Cr, Al, and Si coating components do not completely mix during deposition and form two phases. In this structure, AlTiN and AlCrN nanocrystals are embedded in an amorphous SiN matrix. Silicon increases viscosity and reduces internal residual stress in the coating. The increase in hardness is ensured by a special structure in which the amorphous SiN matrix envelops hard grains and prevents their growth.

It should be noted that the structure of coatings was developed and optimized by the engineers of Platit AG, when our main task was in establishing how the developed coatings built on several different principles and obtained in the conditions of real production would work in the industrial conditions when milling a titanium alloy.

The configurations of the cathodes located in the vacuum chamber of the technological unit during the application of various multi-component coatings on carbide end mills are presented in Figure 4.

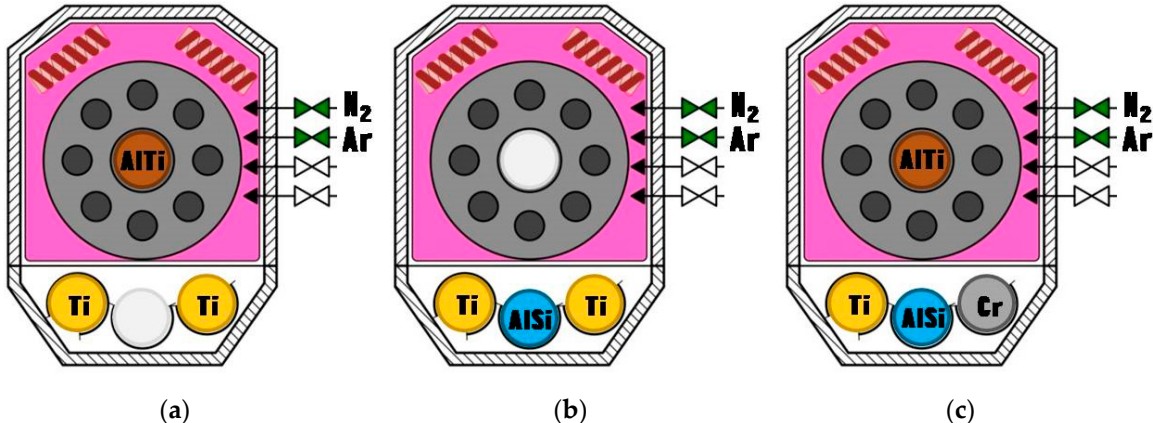

**Figure 4.** The configuration of the cathodes located in the vacuum chamber of the technological unit during the application of various multi-component coatings on carbide end mills: (**a**) TiN–Al/TiN; (**b**) TiN–AlTiN/SiN; (**c**) CrTiN–AlTiN–AlTiCrN/SiN.

The innovative vacuum-arc equipment provides for a different configuration of rotating cathodes depending on the composition and structure of the coating being formed. The cathode that is located on the vertical axis in the center of the vacuum chamber implements the central rotating arc cathode (CERC) technology, and with the lateral location of the covered tool—the already mentioned LARC technology.

Four possible rotating cathodes and a four-channel gas inlet system can be installed in the unit following its design and technological installation. The electron beam between the cathodes generates a dense cloud of highly ionized plasma which cleans the surfaces of the tool, even with the most complex structure.

Preliminary etching of the end mills for pre-treatment operation to clean or etch them from the industrial contaminations before coating was carried out with argon ions with an energy of 500 eV at a pressure of 1 Pa by means of a non-self-contained gas discharge ignited between the cathodes. The electron flow between the two targets creates a high-density plasma in which the products to be processed are immersed. At the same time, a negative bias voltage of 400 V is supplied to the rotary table with the tools being machined. A wear-resistant coating of the selected composition is formed when the stage of cleaning the end mills is completed. The formation of an adhesive sublayer was provided by etching to increase the adhesion of the applied coating with a carbide base in all cases.

Two peripheral Ti-based cathodes and one Al-based central cathode (containing about 10% titanium fraction in the composition) are used when applying the TiN–Al/TiN coating (Figure 4a). A sandwich-type structure is provided by varying currents on titanium and aluminum cathodes.

Three peripheral rotating cathodes are used—two based on Ti and one based on AlSi (the Si fraction is approximately 12%) when applying the TiN–AlTiN/SiN coating (Figure 4b).

One central cathode based on Al and three peripheral cathodes based on Ti, AlSi, and Cr are used in the formation of the CrTiN–AlTiN–AlTiCrN/SiN coating (Figure 4c).

In all cases, the deposition of coatings were carried out in a gas mixture of nitrogen and argon (N—90%; Ar—10%) at a pressure of 1.5 Pa, which gradually decreased to 0.7 Pa. The negative bias voltage of 500 V was applied to the rotary table with cutting tools during deposition, and current on the cathodes varied in the range of 75–120 A (the specific value depends on the technological task to be solved). The coatings were deposited within 120–140 min and provided a total coating thickness of 4.0–4.2 µm.

It should be noted that the deposition rate was 3 µm per hour. However, it substantially depended not only on the geometry of the tool, since the coating may grow more slowly on some surfaces than on others, but mostly on the loading of the working chamber. The dependence between the deposition rate and the composition of the proposed coatings was not identified.

The structure of the surface layer of a carbide tool after applying various multi-component coatings is provided in Figure 5.

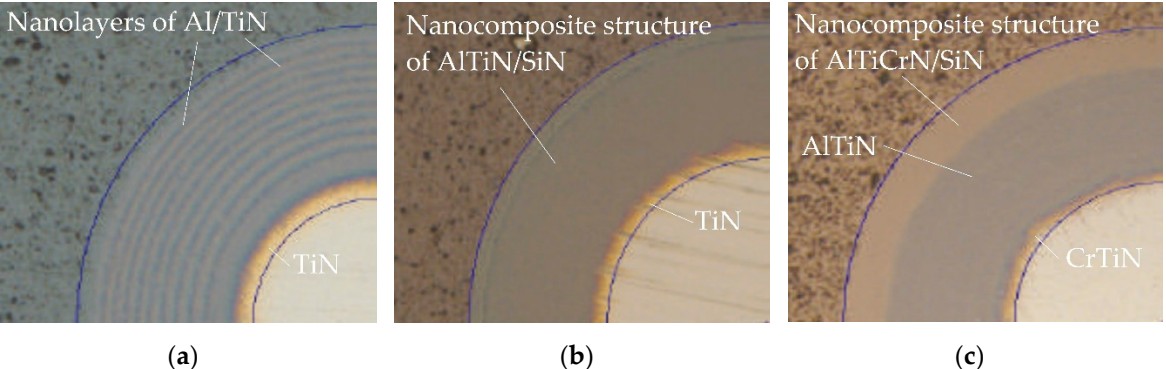

**Figure 5.** Images of the composition number of layers of the formed multi-component coatings of various compositions by analysis on a Calotest instrument: (**a**) TiN–Al/TiN; (**b**) TiN–AlTiN/SiN; (**c**) CrTiN–AlTiN–AlTiCrN/SiN.

It should be emphasized that etching and pre-treatment of the cutting tool plays a key role in the coating adhesion. In the considered case, an industrially produced end mill was already ground in the manufacturing conditions. Then it was properly etched and coated by Platit technology with the developed multi-component coatings that provided blunted edges of the tool to stabilize the cutting edge since the sharp edge of carbide tool under the influence of high loads is destroyed and quickly wears out. As it is known, a simple blunting of the cutting edge to about 15 μm can significantly stabilize the cutting edge, as well as significantly increase the service life and reliability of the cutting tool. However, the used carbide tool was blunted by coating. The cutting edge in the considered conditions should be not pre-blunted mechanically before coating; otherwise, the obtained radius of cutting edge after coating will not allow us to cut anything. The adhesion of the coatings was proved further by the conducted tests.

### 2.3. Characterization and Properties of the Samples

Sample preparation was carried out according to standard metallography techniques. The thickness and number of layers of deposited coatings was controlled by a Calotest instrument from CSM Instruments (Freiburg-im-Breisgau, Germany).

An analysis of the morphology and surface topography of carbide tool samples with various multi-component coatings was carried out using a scanning electron microscope VEGA3LMH (Tescan, Brno, The Czech Republic) and a high-precision profilograph–profilometer HOMMEL TESTER T8000 (JENOPTIK Industrial Metrology Germany GmbH, Jena, Germany). The radius of the rounding of the cutting edge was controlled using an optical 3D measuring system MicroCAD premium+ (GFMesstechnik GmbH, Teltow, Germany).

It is known that the correct measurement of the hardness and modulus of elasticity of thin coatings is possible using precision equipment that allows providing indentations with contact depths less than the film thickness (10–20%) [64]. The use of special techniques and standards that can complement each other results in increasing the accuracy of the measurement. A nanoindentation scheme was used by the Berkovich diamond indenter during the experiments. That scheme is more reliable in contrast to the widespread Vickers pyramid microhardness measurement scheme. The nanoindentation scheme eliminates the influence of the substrate hardness on the measurement results and establishes the required penetration depth of the indenter when measuring the hardness of a thin film.

The CSM instruments nano hardness tester (Freiburg-im-Breisgau, Germany), which made it possible to provide a penetration depth of 0.1–0.3 μm, was used in the experiment. Unloading began after the load on the coated sample reached the set value. In other words, the load acting on the indenter

gradually decreased to zero. The microhardness load was 0.25 N; the duration of the load–unloading cycle was 50 s. The quantitative value of hardness was determined by certified software based on an algorithm proposed by W.C. Oliver and G.M. Pharr [65–67].

An analysis of indentation curves makes it possible to determine not only the hardness *H* but also the elastic modulus *E*, the ratio of the total indentation work to its elastic and inelastic (plastic) components. The *H/E* ratio is called the plasticity index, and its quantitative value allows you to indirectly judge the viscosity of the coating and its ability to resist deformation. The noted property is extremely important when operating the coating in the conditions of milling titanium alloys with increased heat and power loads. Previous works [68,69] have shown that the material should have high hardness with a reduced modulus of elasticity to increase the resistance of the material against elastic fracture and reduce plastic deformation.

## 2.4. Adhesion and Wear Tests of End Mills

Obviously, the effect that is expected from coating the working surfaces of the tool is not only to increase the microhardness of the contact pads in contact with the material being processed, but also to reduce its adhesion to the tool. When processing titanium alloys, it is especially important to choose a coating that reduces the rate of sticking of the processed material to the cutting tool. This is due to special properties of titanium-based alloys. Increased power and thermal loads in the cutting zone accompany their machining. As a result, heat removal from the cutting zone is difficult, and due to the strong adhesive interaction, titanium particles adhere to the working surfaces of the tool. It increases the wear rate of end mills.

The process of titanium sticking to end mills was studied by means of high-quality SEM analysis of the distribution of chemical elements on their front surface.

The wear tests were carried out on a CTX beta 1250 TC 4AB turning and milling machining center (DMG MORI, Bielefeld, Germany) equipped with Siemens SINUMERIK 840D numerical control system with ShopTurn 3G software. In the process of testing end mills, the strategy of trochoidal processing of a rectangular groove of 20 mm in a titanium alloy workpiece was implemented. A titanium alloy for aviation purposes was used as a material to be processed. The chemical composition of which is given in Table 1. It should be noted that the titanium workpiece was in the delivery state with a hardness is of 318 HB.

**Table 1.** Chemical composition (%) of the titanium alloy to be processed.

| Ti | Al | Zr | V | Mo | Si | N | Fe | C | H | O |
|---|---|---|---|---|---|---|---|---|---|---|
| 85.15–91.4 | 5.5–7 | 1.5–2.5 | 0.8–2.5 | 0.5–2 | Up to 0.15 | Up to 0.05 | Up to 0.25 | Up to 0.1 | Up to 0.015 | Up to 0.15 |

The thermal shock and heating are the aspects that should be taken into account in the development of the cutting mode as the release of a large amount of heat during milling of heat-resistant and titanium alloys limits the cutting speed. The application of the trochoidal milling strategy allows redistributing and reducing cutting loads by providing the constant small cutting curve and, consequently, a constant width of a chip when the radial depth of cut $a_e$ is variable. The strategy allows involving a longer cutting part of the end mill in the process and removing a large volume of the material due to the more even loads. The particular geometry of the tool also assists in obtaining finer chips and constant tool loads in axial and radial directions with an effective number of teeth involved in cutting. It is also recommended using the coolant in hard-to-machine materials machining for easier heat and chip removal, avoiding secondary involving chips in the process of cutting. However, tests of new materials and/or coatings should always be carried out without coolant to exclude the influence of coolant on wear resistance and evaluate the contribution of the coating since the coolant itself increases tool wear resistance, affects contact friction and interaction.

In the experiments, the coated end mills were tested without coolant to provide as well better observation of wear and titanium adhesion on cutting edges, and that correlates to the standard tool tests when the used strategy should allow reducing in cutting loads and consequently reducing in thermal shock and heating. The trajectory of trochoidal milling was quite sophisticated, and the chosen parameters were corresponding to this condition: cutting speed ($V$) of 131 m/min; rotational speed ($n$) of 3500 mm/rev; tooth feed ($f_z$) of 0.094 mm/tooth; max radial depth of cut ($a_e$) of 2.4 mm; axial depth of cut ($a_p$) 5 mm; feed rate ($V_s$) 1200 mm/min. The recommended axial depth of cut is recommended to choose in the range of $1 \div 1.5 \times D$ (up to $2 \times D$) where $D$ is a mill diameter; however, a smaller value of axial depth of cut $a_p$ was taken to reduce cutting loads and thermal shock with enlarged tooth feed $f_z$ and cutting speed $V_c$.

The criterion for tool failure was the achievement of wear ($h_f$) along the rear surface of the teeth on the cylindrical and on the front parts of the milling cutters of a quantitative value of 0.3 mm. The cutting path traversed by the tool to achieve the specified wear was adopted as the wear resistance of the end mill. The quantitative value of wear was repeatedly evaluated using a SteREO Carl Zeiss instrumental stereomicroscope (Carl Zeiss Microscopy, Jena, Germany) after the tool passed a cutting path of 100 m.

## 3. Results

### 3.1. Surface Morphology and Microrelief

The SEM images of the surface of hard alloy samples obtained at 10,000× and 500× magnifications are shown in Figure 6. It can be seen that the surface of the tool before the coating had characteristic grooves that formed during diamond grinding and sharpening of the tool (Figure 6a).

The images after coating deposition (Figure 6b–d) demonstrate that coatings were formed with a minimum content of the micro-droplet component. There, it can be observed aluminum droplets only on the surface of the TiN–Al/TiN coating (Figure 6b). However, a number of these droplets are unambiguously smaller than the researchers observed in traditional cathodic arc technologies during the synthesis of aluminum-containing nitride films.

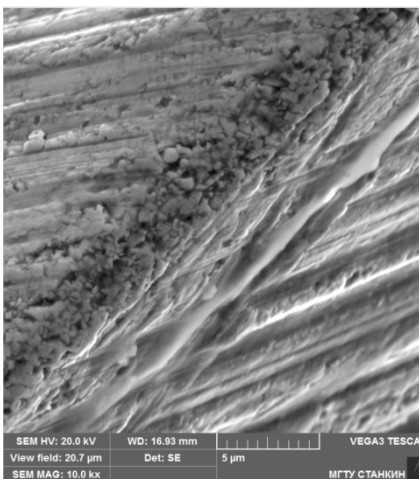 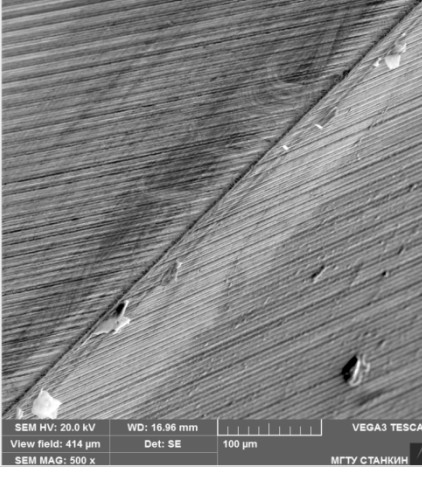

(**a**)

**Figure 6.** *Cont.*

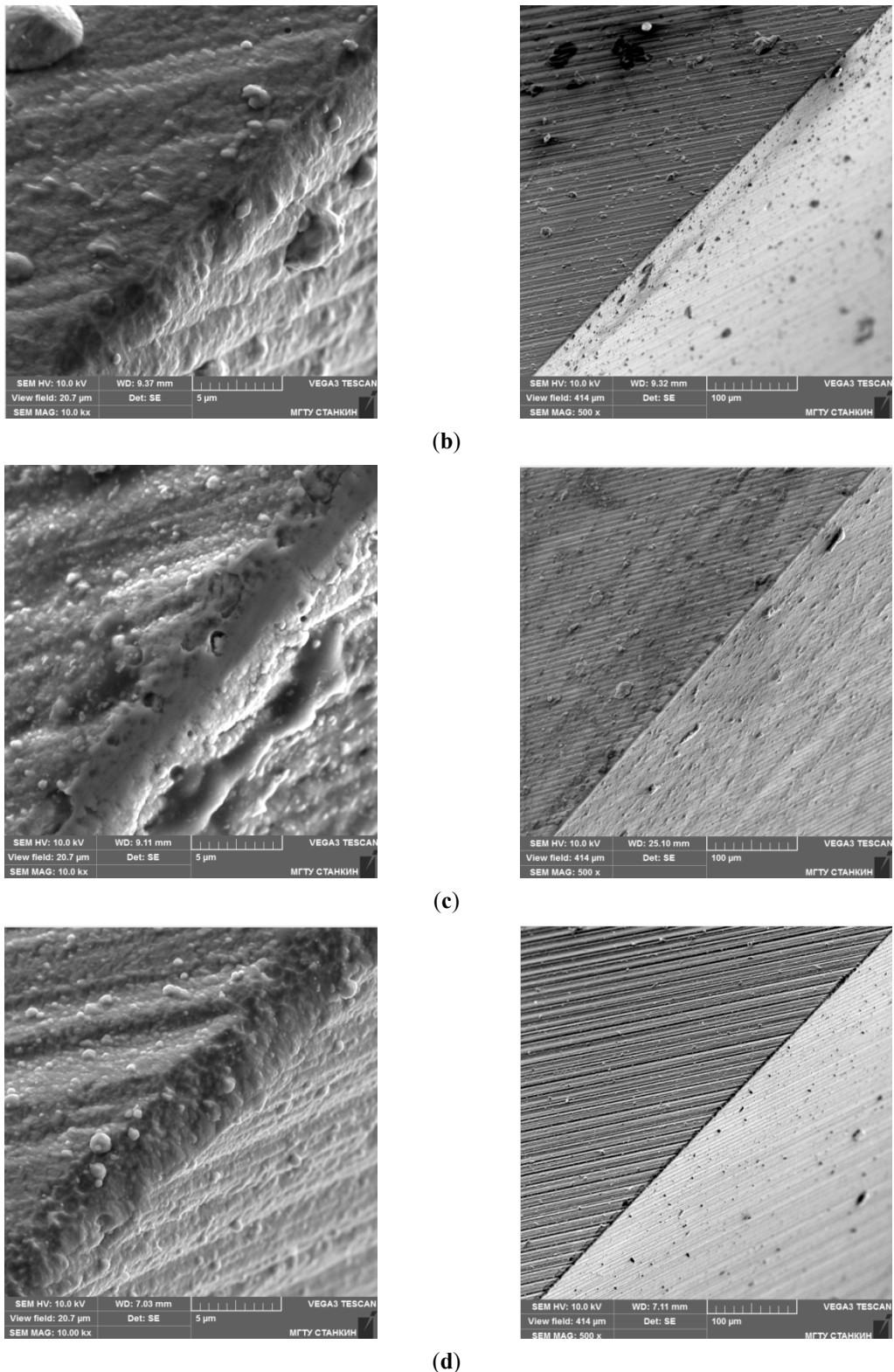

**Figure 6.** SEM images of the surface and cutting edge of carbide end mills before and after applying various multi-component coatings with a magnification of 5000× in the left column and 500× in the right column: (**a**) uncoated; (**b**) TiN–Al/TiN; (**c**) TiN–AlTiN/SiN; (**d**) CrTiN–AlTiN–AlTiCrN/SiN.

In this case, the microrelief (roughness) of the surface (Figure 7a) has a classical appearance with a quantitative value of 0.3–0.7 μm. After applying the cathodic arc coatings, the morphology

of their surface and the microrelief noticeably change. For example, the grooves from the previous grinding were smoothed, but microdrops 0.5–1 μm in size were found on the surface after applying a TiN–Al/TiN coating (Figure 6b). Analysis of the 3D profilogram (Figure 7b) allows us to see these particles in the form of characteristic peaks with heights of about 1 μm. The TiN–AlTiN/SiN coating (Figure 6c) was insignificant, but it still affected the surface morphology. A smoother relief was clearly visible in the profilograms (Figure 7c). The coating fills the cavities and recesses remaining after grinding. Moreover, the quantitative roughness value was 0.3–0.7 μm, which is similar to the tool before coating. The smoothed surface can be visually observed in the CrTiN–AlTiN–AlTiCrN/SiN coating tool (Figure 6d), which also confirms the nature of the profilogram (Figure 7d) that the surface looks uniform with a roughness of 0.3–0.6 μm.

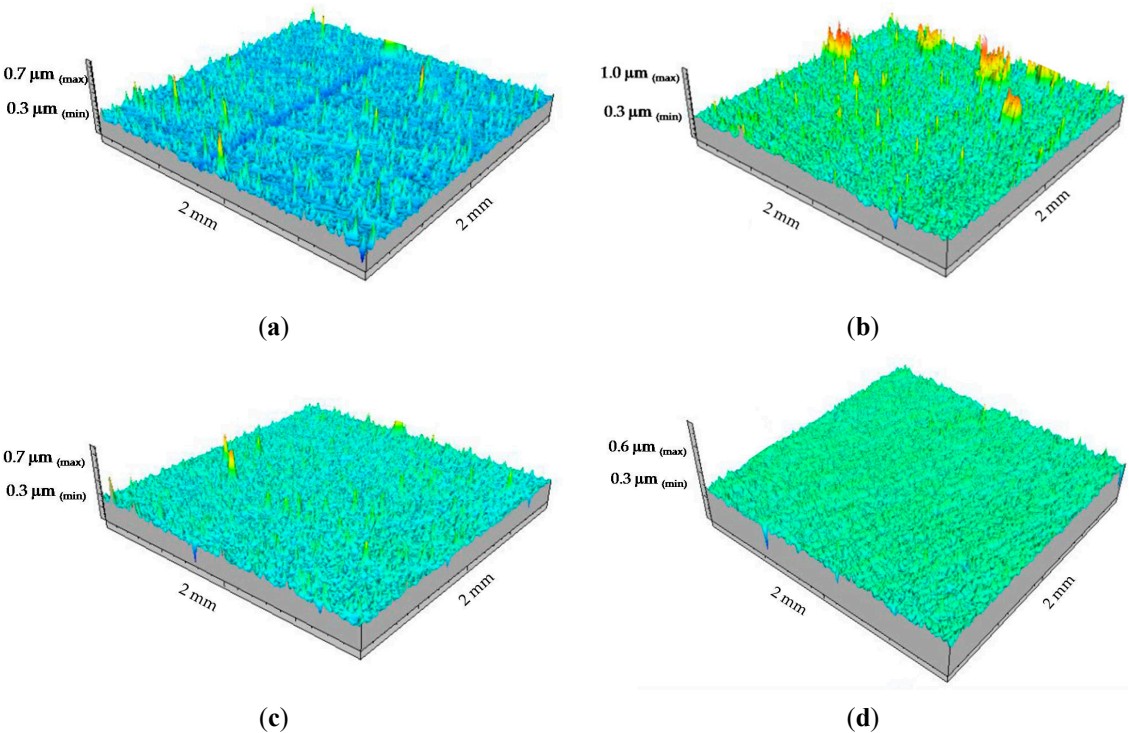

**Figure 7.** Microrelief of the surface of carbide end mills before and after applying various multi-component coatings (images obtained using an automated profilograph–profilometer): (**a**) uncoated; (**b**) TiN–Al/TiN; (**c**) TiN–AlTiN/SiN; (**d**) CrTiN–AlTiN–AlTiCrN/SiN.

It should be noted that the measured radius of the rounding of the ground and sharpened cutting edge was around ~6–8 μm for untreated cutting tools, and ~18–22 μm for coated end mills.

*3.2. Physical and Mechanical Properties*

In the course of further studies, the physical and mechanical properties of multi-component coatings were evaluated. Table 2 presents the results of complex measurements. It can be seen that all the coatings under study were characterized by a satisfactory plasticity index (*H/E* ratio) and showed a value close to 0.1. The maximum value of this indicator (0.12) was demonstrated by the CrTiN–AlTiN–AlTiCrN/SiN coating.

**Table 2.** Physical and mechanical properties of cathodic arc deposited multi-component coatings.

| No. | Multi-Component Coating Composition | Microhardness H (GPa) | Elastic Modulus E (GPa) | H/E Ratio |
|---|---|---|---|---|
| 1 | TiN–Al/TiN (multilayered "sandwich") | 33 ± 1 | 362 ± 8 | 0.091 |
| 2 | TiN–AlTiN/SiN (nanocomposite) | 43 ± 2 | 412 ± 6 | 0.104 |
| 3 | CrTiN–AlTiN–AlTiCrN/SiN (multilayered nanocomposite) | 41 ± 1 | 340 ± 6 | 0.120 |

### 3.3. Adhesion and Wear Resistance of End Mills

Before the analysis, all tools went through a 200 m cutting path and were worn out because of contact with a titanium alloy workpiece. The contrasting images obtained using SEM analysis tool are presented in Figure 8, where titanium particles are displayed in yellow color.

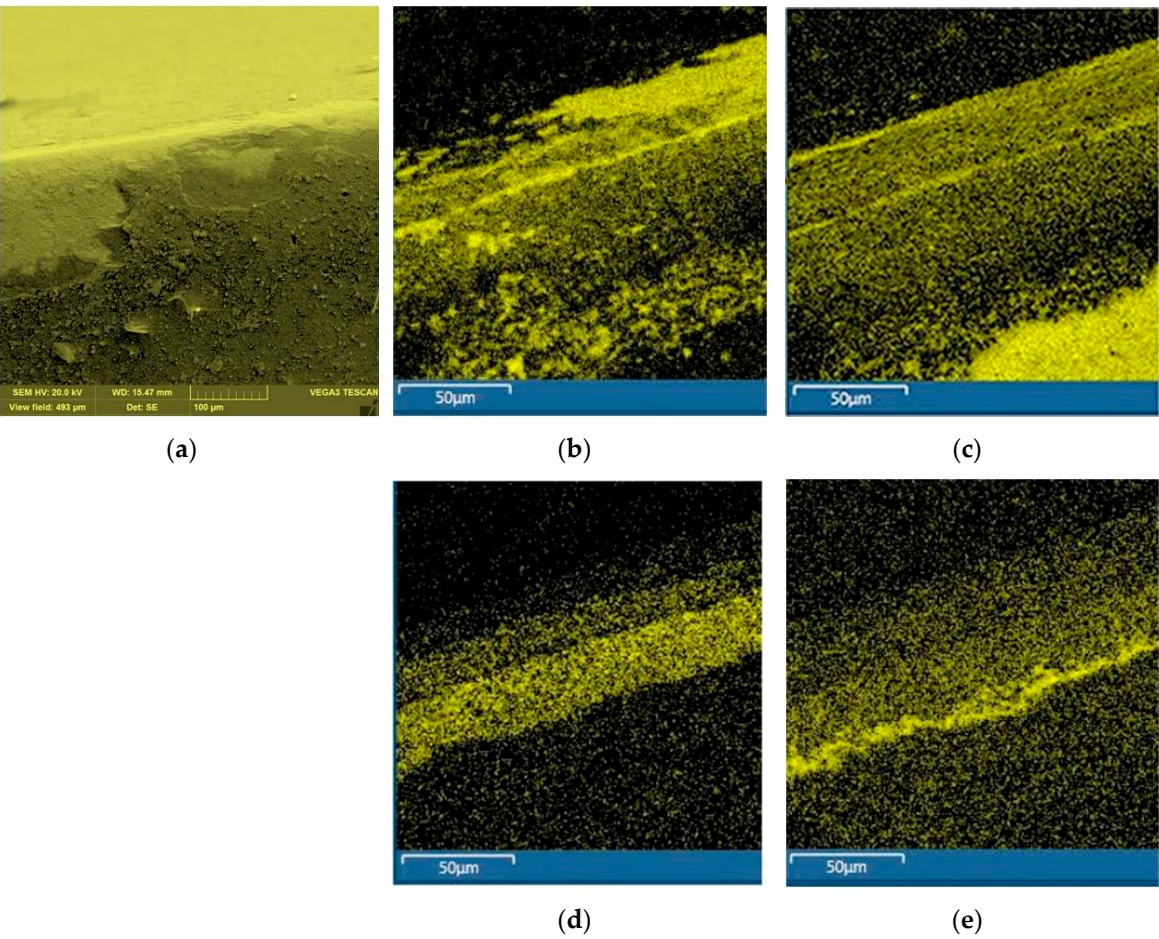

**Figure 8.** Contrasting SEM images of the distribution of titanium (yellow particles) on the front surface of end mills with different coatings after passing a cutting path of 200 m: (**a**) image of the front surface of the tool; (**b**) uncoated; (**c**) TiN–Al/TiN; (**d**) TiN–AlTiN/SiN; (**e**) CrTiN–AlTiN–AlTiCrN/SiN.

The presented data demonstrates well that in the case of using a tool without coating, intense adhesion of titanium was observed (Figure 8b). The TiN–Al/TiN coating (Figure 8c) did not make a significant contribution to the decrease in the rate of titanium sticking to the front surface of the cutters during operation. The opposite picture is observed during the operation of a tool with TiN–AlTiN/SiN coatings (Figure 8d) and CrTiN–AlTiN–AlTiCrN/SiN coatings (Figure 8e). One can observe a pronounced positive effect—the number of titanium particles decreased sharply, an ordered

array of small particles of titanium was present on the tool: this allows us to conclude that the intensity of titanium sticking to the tool was reduced.

The results of operational tests of end mills with various coatings with a trochoidal strategy for processing a groove in a titanium alloy workpiece are shown in Figure 9. The obtained graphs demonstrated that the "wear–cutting path" relationships had a traditional trend. At the first stage (up to 100 m), there was a "running-in" of tool and processed materials along the back surface. Then followed the stage of delayed (stable) wear. At the last stage of operation, wear was noticeably intensified, the surface being machined becomes extremely rough, and the cutter fails. A huge difference and a significant effect from many coatings under study were seen when we analyzed the quantitative value of wear of end mills at the time of failure (the wear reached 0.3 mm). The failure of endless milling cutters occurs after a cutting path of 340 m, for a tool with a TiN–Al/TiN coating—1100 m, for a tool with a TiN–AlTiN/SiN coating—1380 m, and for a tool with a CrTiN–AlTiN–AlTiCrN/SiN coating—1580 m.

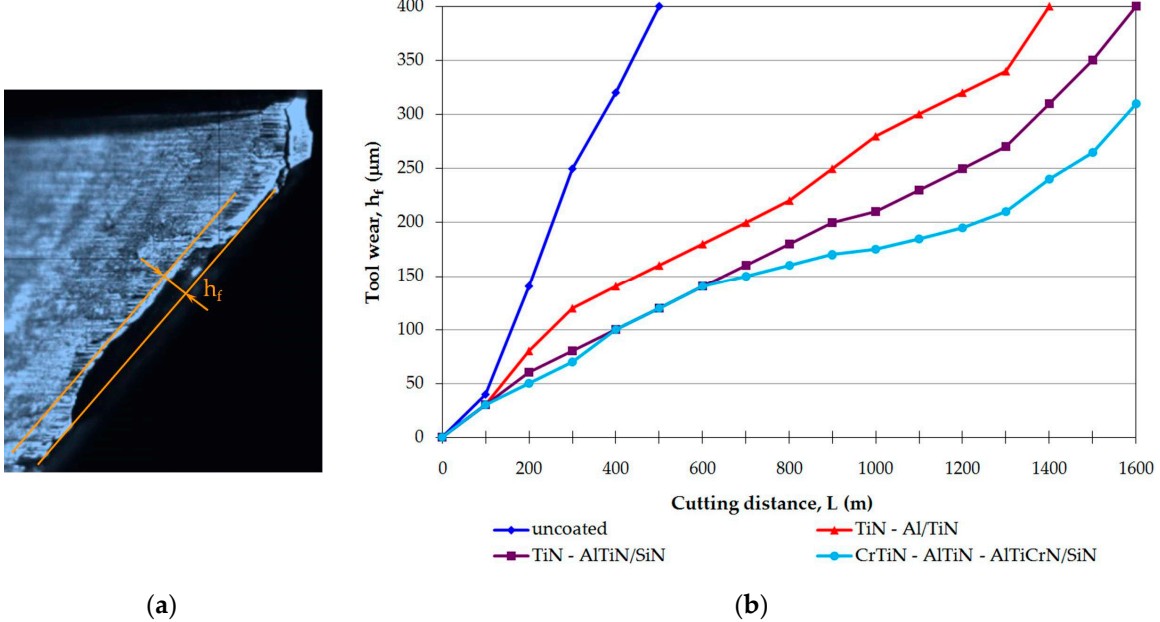

(**a**)　　　　　　　　　　　　　　　　　　　　(**b**)

**Figure 9.** The scheme for measuring wear of the end mill and the "wear – cutting path" relationship of end mills with different coatings with a trochoidal strategy for machining a rectangular groove in a titanium workpiece in the following modes: of $V = 131$ m/min; $n = 3500$ mm/rev; $f_z = 0.094$ mm/tooth; $a_e = 2.4$ mm; $a_p = 5$ mm; $V_s = 1200$ mm/min: (**a**) microphotograph; (**b**) graphs.

## 4. Discussion

The complex of applied research shows how multi-component cathodic arc coatings TiN–Al/TiN, TiN–AlTiN/SiN, and CrTiN–AlTiN–AlTiCrN/SiN affect the properties of the surface and subsurface layer of carbide end mills and their wear resistance with a trochoidal strategy of milling titanium alloy.

The effect of TiN–AlTiN/SiN and CrTiN–AlTiN–AlTiCrN/SiN coatings on the morphology and surface roughness of a 6WH10F carbide tool was not very significant. A smoother relief was noted since the coating slightly filled the cavities and recesses remaining after grinding, but the maximum roughness value was 0.3–0.7 μm, i.e., similar to the tool before coating. After coating with TiN–Al/TiN, characteristic peaks of up to 1 μm were found on the surface. This is apparently due to the small microparticles up to 1 μm in size that were formed during the deposition of coatings, which were detected by SEM analysis.

The choice of the composition of a multi-component coating for carbide end mills processing titanium alloys is not an easily solved problem [70,71]. Before taking a choice on composition and

structure of the protective coating, plenty of the process parameters should be taken into account. Surface roughness, residual stress, microstructure, and hardness of the machined surface can be called the primary parameters of the workpiece to be machined. The deposed coatings have a role of protective film for cutting edge geometry that provides a decrease in friction coefficient and cutting temperatures and improves the performance and quality of product surface.

The chosen coatings are related to the last achievements in the field of protective coatings development, that previously were monolayered but current studies proved that deposited on the substrate two- or even three phase structured multi-component and nanocomposite coatings have brighter pronounced protective effects on main exploitation properties of cutting tools [72].

Nitride coatings for cutting tools based on TiN, TiAlN, CrN, ZrN, TiSiN, TiAlSiN, CrAlN, TiAlCrN, and cBN systems are widespread and received a development due to the fact of its inertness to the environment and machining surfaces, excellent adhesive properties to the tool, proper combination of the exploitation characteristics shown during milling the most hard-to-machine materials especially such a specific material as titanium alloy [73,74]. The nitride-coated tools showed overall better performance, surface finish quality, and performed geometry of the product in comparison with uncoated tools [75]. It is proved that some nitride coatings as AlCrN and TiAlN show greater hardness and wear resistance than for example TiAlSiN coated tools [76]. Meanwhile, the TiAlN coating shows the better surface quality and geometric accuracy of the products.

It should be noted that the proposed composition and structure of the coatings in combination with the researched milling strategy has no analogs between previously conducted studies but complete the overview of the research domain to make wider and precise for the industrial applications.

In this context, the researchers and industry professionals should always ground their choice on the purpose of the coating: improved performance or finishing of the machined surfaces. With the whole complex of the already conducted researches and experimental data, it is possible to find suitable composition and structure of the coating for each purpose regarding the used workpiece material, milling strategy, and main cutting parameters, chosen cutting tool geometry.

We were convinced in the course of our experiments that maximum hardness and elastic modulus do not mean maximum wear resistance at all. The coating must have high hardness with a reduced modulus of elasticity to resist high power and thermal loads during milling of titanium alloys. The coating of the end mill must have a high plasticity index in order to resist abrasive and adhesive wear simultaneously and did not deform the surface layer during machining. In our case, the maximum value of this indicator at the level of 0.12 was shown by the CrTiN–AlTiN–AlTiCrN/SiN coating. In addition to this property, the critical role is played by the coating's tribological characteristics, which can be indirectly estimated by the rate of adhesion and welding of titanium particles to the working surfaces of the end mills during the processing of the titanium alloy.

The experiments demonstrated that there is a fairly intense adhesion of titanium during cutting when using the tool without coating and using the TiN–Al/TiN coated tool. At the same time, we observed a pronounced positive effect in the tool with TiN–AlTiN/SiN and CrTiN–AlTiN–AlTiCrN/SiN coatings when the amount of adhering titanium particles was sharply reduced, which allowed us to conclude that the adhesion intensity was reduced.

The obtained results showed that multilayer and nanocomposite structures with the proposed content of the coatings provides improved wear resistance of coatings and reduced titanium adhesion in comparison with the uncoated tool. At the same time, it is evident that the complexity of the coating plays a key role in their resistance since its structure hampers the development of cracks that cannot be developed through the structure of the coating due to the designed obstacles in the form of multiple layers of coating and nanocrystals in an amorphous matrix of the nanocomposite. It correlates to the hypothesis that, during wear, the obtained nanocomposite and multilayered structures of coatings provide changes in the direction of cracks, and it hampers of crack development that has a noticeable impact on wear resistance of the coatings.

## 5. Conclusions

It can be concluded that TiN–Al/TiN is not an optimal choice for working under these processing conditions, even considering that all the coatings studied made a significant contribution to increasing the tool wear resistance during the operational tests. This is most likely due to the higher affinity of this coating with a titanium alloy and lower ductility index compared to the other two coatings studied.

The results of testing tools with coatings with a trochoidal milling strategy for titanium alloys show that all studied coatings had a positive effect on the course of the cutting process, change the nature of the contact interaction between the tool and the workpiece, performing the function of a solid lubricant and reducing the wear rate. The maximum effect was achieved when using a tool with a CrTiN–AlTiN–AlTiCrN/SiN coating—end milling time (wear resistance) was increased 4.6 times compared with uncoated samples, 1.4 times compared to TiN–Al/TiN coating and 1.15 times compared to TiN–AlTiN/SiN coating.

The provided experiments in the conditions of titanium alloy trochoidal milling showed that the complexity of the coating structure improves its wear resistance that can be related to the changes in crack development that also become more sophisticated by the complexity of the coating.

It should be emphasized that the practical significance of the work is in using trochoidal milling strategy, which has never been researched before in the context of end milling titanium alloys with a PVD-coated tool when there are plenty of published works related to the use of other traditional milling strategies. Particular interest in such work can arise not only among scientists in the subject field but also among the real industry sector at various levels since the research on the behavior of the proposed coatings under the conditions of efficient trochoidal milling was almost never studied before but stayed in demand. Moreover, the structure of the proposed and developed coatings corresponds to the recent advances in thin coatings of the last generation.

**Author Contributions:** Conceptualization, M.A.V.; Methodology, S.V.F.; Software, M.M.; Validation, S.O.; Formal Analysis, M.A.V.; Investigation, S.V.F.; Resources, S.O.; Data Curation, M.M.; Writing—Original Draft Preparation, S.V.F.; Writing—Review and Editing, M.A.V.; Visualization, S.O., M.M.; Supervision, S.V.F.; Project Administration, S.V.F.; Funding Acquisition, M.A.V. All authors have read and agreed to the published version of the manuscript

**Funding:** This project has received funding from the Ministry of Education and Science of the Russian Federation within the framework of the state task for scientific research, under Grant Agreement No. 0707-2020-0025.

**Acknowledgments:** The research was done at the Department of High-Efficiency Processing Technologies of MSTU Stankin.

**Conflicts of Interest:** The authors declare no conflict of interest.

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
