# Peer review of "Wear Resistance and Titanium Adhesion of Cathodic Arc Deposited Multi-Component Coatings for Carbide End Mills at the Trochoidal Milling of Titanium Alloy"

_technologies, doi:10.3390/technologies8030038_

Round 1

Reviewer 1 Report

The aim of the paper „The effectiveness of cathodic-arc deposited multicomponent coatings for carbide end mills at the trochoidal strategy of machining titanium alloy“, is quite interesting. Results published in this article have a sufficient impact and add to the knowledge base. The aim of the paper is in good agreement with expectations and focus of the journal. The paper is written in good English.The references are appropriate and up to date. In conclusion, I recommend this paper for publication with minor revision:

line 2: instead of „deposed“ write „deposited“

line 15: instead of „...microscopy: The surface roughness was estimated.“ write „...microscopy. Surface roughness was estimated.“

line 15: instead of „The microhardness...“ write „Microhardness...“

line 17: instead of „titanium sticking“ write „sticking titanium“

line 17: instead of „to the tool working surfaces“ write „to the working surface of the tool“

line 22: instead of „compared to with“ write „compared with“lineR23: same as R 22

R 39: instead of „machining stragegy, improvement of“ write „machining strategy, such as improvement of“

line 40: omit „with“

line 43: instead of „processing titanium alloy“ write „processing of titanium alloy“

line 64: instead of „of the technological system "machine tool - fixture - tool - part" write „of the „machine tool – fixture – tool – part“ technological system“

line 72: instead „increased to 6-8 times“ write „increased 6-8 times“

lines 86 and 87: omit „a“ before words „diameter“, „length“, cutting part“ and „spiral angle“

lines 88 and 89: omit „a“ before words „hardness“ and „density“

line 101: omit a comma before word „leads“

line 113: instead of „characteristic of“ write „characteristic for“

line 114: instead of „deposed“ write „deposited“

line 117: omit „The“

line 122: instead of „processing titanium alloy.“ write „processing of titanium alloy.“

Line 138: The Calotest is used to measure the thickness of the coating. Also to determine the number of layers. Therefore, I recommend: insread “The microstructure of the deposited coatings” write “The thickness and number of layers of deposited coatings”

line 153: instead of „, was used in the experiment“ write „were used in the experiment“

line 158: omit „the“ in front of „indentation curves“

line 163: instead of „when“ write „with“ omit „act“ at the end of the sentence

line 165: instead of „the resistance of the material to elastic fracture“ write „the resistance of the material against elastic fracture“

line 171: instead of „because of the“ write „due to“ omit „the“ in front of „special properties“

line 226: instead of „deposing“ write „deposition

line 231: insteas „microstructure“ write „composition number of layers“

line 262: instead of „deposed“ write „deposited“

line 267: instead of „where the yellow particles are titanium particles“ use „where titanium particles are displayed in yellow color.“

line 272: instead of „demonstrate“ write „demonstrates“

line 312: instead of „do not have deformed“ write „did not deform“

line 332: omit „by“

Page 6, figure 4a,b,c: I recommend to mark the top layer and the bottom layer

The Conclusions section is missing. Autors must complete Conclusion.

Reviewer 2 Report

Dear Authors,

Congratulations on your work, which is in line with the research I'm leading right now. Maybe due to that, my criticism is particularly high regarding your work, but I will provide some suggestions and comments in order to improve your work, allowing that it can be considered for publication after those amendments. My concerns are only in technical and scientific terms, because the paper is very well written, showin the experience of some Authors in publishing goo papers in the near past. Thus, please pay attention to my comments and suggestions, hoping that they can be of interest for you:

  1. The abstract, as well as all work shown, should be focused on the wear of the tools. Some dispersion is felt, namely through too much references to the deposition process, mainly the reactor deposition technique. This should be described in the Methods section.
  2. Keywords don't refer the machining process. I think it would be important.
  3. The Introdution is not clear and is focused in large quantities of references for each information given. Moreover, it is not focused in the previous work carried out by several Researchers in this field of knowledge. The Introduction is too much centered in self references or references from the same group of work (Prof. Volosova, Prof. Grigoriev, Prof. Fyodorov, etc.). I recommend to include other recent references such as: https://doi.org/10.3390/coatings8110402;   https://doi.org/10.3390/ma12020233; https://doi.org/10.3390/coatings8020059; https://doi.org/10.3390/coatings7080127;  https://doi.org/10.3390/met8100850; among others.
  4. The writting process shouls be direct: Cheng et al. [XX] studied the ......using .....methodology, concluding that....
  5. The last sentence in Page 1 that ends in Page 2 needs a reference.
  6. More references should be added regarding the type of PVD coatings used in this work (or similar).
  7. At the end of the Introduction, a framework regarding the structure of the paper should be presented.
  8. No previous studies about machining process of Ti alloys are referred.
  9. Please point out the cutting edge radius of the tool;
  10. References 29-42 are pointed out in the Methods section. This reveals poor organization of the paper, because the inclusion of references into the Methods section is usual when we need to refer to procedures or equipments used in other works. Please revise this situation.
  11. The section 2.1 is too much focused on Deposition process. This is not the main focus of your work. Thus, please just provide the type of reactor and the deposition conditions used in your work, not describing in detail the reactor, plasma and deposition process.
  12. Please point out the deposition rate for each one of the conditions used.
  13. Please explain whar do you mean with the following sentence in 2,2: "...of coatings formation for instrumental purposes". Instrumental purposes?
  14. Please also explain the sentence in page 3: "...multilayer coating has a higher viscosity...".
  15. Please also explain the sentence in page 4: "...is able to absorb microcracks between the layers....". Could you explain the formation of these microcracks? How they evolve? Could you refer the main causes? Residual stresses?
  16. The reasons for the selection of the three coating architectures is not properly explained. The readers would be happy if you could provide information about the main reasons behind these options, maybe through previous work.
  17. In line 145, please explain the following sentence: "...that allows for indentation at depths less than the thickness of the film.".
  18. Please point out the load used in the microhardness measurements.
  19. Please care the units of cutting speed in line 185.
  20. Please be sure about the feed rate: 1200 mm/min. 
  21. In page 198-202, the novelty of the equipment is not clearly stated. 
  22. Are the conditions homogeneous inside the whole reactor?
  23. In line 210, when you refer "...negative potential of 400 V...", are you wishing to refer "bias"?
  24. Coud you provide figures about the films' thickness?
  25. Why it is not provided accurate information (AFM) about the coatings surface roughness? Figure 6d shows significant roughness, which should be higher than the range provided to the reader (0.3 - 0.7 um).
  26. In figure 8, please provide information about the coating wear and substrate wear, because it is clear that the substrate is also affected.
  27. The Conclusions are mixed with the discussion.
  28. The Discussion should compare your results with the results achieved by other Researchers (5, at least).
  29. The Conclusions should be aligned with the main goals of the paper.
  30. The title seems do not correspond to the work. Please revise.

Hope these ideas can help you improving your paper.

Kind regards,

Reviewer

Reviewer 3 Report

  • First of all the General agreement of international societies about coating tools, is that stop works in which edge preparation is not treated properly, being one of the most key parameter for coating performance. Bouzakis and CIRP were clear about that.
  • Coatings is not only a complete strand in nanotechnology, the success of a layer is to rest on a pre-treated surface and to receive some post treatment as well. Results can change dramatically depending on one or other “craft” in addition to science. Few references but important are DOI:10.1080/10426914.2014.973582, regarding the ways to get more quality of surfaces and tool or other application performance. The work followed that of other authors: https://doi.org/10.1016/j.jmapro.2017.01.012 and The Int. J. of Advanced Manufacturing Technology, July 2014, Vol. 73, Issue 5-8, pp 1119-1132. An Procedia, Behaviour of PVD coatings in the turning of austenitic stainless steels, Procedia Engineering 63, 133-141, https://doi.org/10.1016/j.proeng.2013.08.241
  • Drag grinding or blasting, pre and post treatment must be discussed.
  • Give some conclusions as bullets, one per each hightligth.
  • Figure 6: are there droplets?
  • A titanium alloy for aviation purposes was used as a material to be processed: aged or annealed, give hardness please.
  • Cutting tool is OK
  • Behavior of austenitic stainless steels at high speed turning using specific force coefficients, The International Journal of Advanced Manufacturing Technology 62 (5-8), 505-515 they showed that cutting speed can change a lot the coating performance.

I expect at least 5-7 better references. More you can find in Polvorosa, Suarez, they really worked hard with turning in one work in Proc inst mechanical engineers, Part C, it was about turning.

Now in USA we are in this lines of thinking, please follow in further research as well.

Round 2

Reviewer 1 Report

Thank you for adding and correcting your article. I wish the authors of the article success in research into the use of thin films/coatings in machining.

Author Response

Dear Reviewer,

Thank you for your kind evaluation of our work and help in improving the manuscript. The last time we have forgotten to revise Figure 4 by your suggestion, now it is corrected (now it is Figure 5). 

Kind regards,

Authors

Reviewer 2 Report

Dear Authors,

Thank you so much for addressing all my comments and suggestions.

Now, the paper is much more clear and informative.

Congratulations.

Kind regards,

Reviewer

Author Response

Dear Reviewer,

Thank you once again for your kind evaluation of our work and help in improving the manuscript.

Kind regards,

Authors 

Reviewer 3 Report

I have read paper and asnwers, and it my regret to inform you that paper did not improve. Trochoidal milling is only a type of cutting without specific relation to coating behavior:

  • Axial depth of cut is high
  • Thermal shock and heating can be key aspects

Tool edge must be tough and the main cutting tool brands (Mitsubishi or Emuge Franken) offer all kinds of information about the process. CAM, programming is the main topic, and operation is reduced to slotting. My experience with CFRP in which we used trochoidal was that. Omsrud is making tools using the approaches mentioned.

This aspect in the paper has not relevance, even the title could change.

The long answer is weird. Really tool edge treatment is more important than the coating recipe itself. I have checked the works, all used blasting or OTEC drag grinding. At the States OTEC is not well-known, but similar are in all cutting tool small companies, around Ohio e lot. You did not mention the real challenge, this, and so the paper cannot be accepted in the present form. About 40-505 successful ratios depend on it.  Looking at Platit website, they really make some offers of machines in this line. Some works inside the long argue are better aimed at the rebuttal because they mentioned more aspects of the entire technique.

My opinion is that the paper could be improved much more with some information from manufacturing journals. You can rewrite some parts of the paper, and it can be accepted in manufacturing journals. Technologies and trochoidal milling (trochoidal is a little idea) are not a perfect match.

Summarizing: if the paper would be improved and better discussed the technical aspects using information from results and by others, could be acceptable.

Author Response

Response to Reviewer 3 Comments, Round 2

Dear reviewer,

Thank you that you take the time to read our manuscript. Unfortunately, we were surprised to know that you found our manuscript not revised from the previous version as we have introduced more than 70 changes and answered in total to 73 points of 3 reviewers. We hope that it was a technical mistake as our revised version was significantly improved from the firstly submitted version and included a few important points that were proposed in your review.

Kind regards,

Authors.

Point 1: I have read paper and asnwers, and it my regret to inform you that paper did not improve. Trochoidal milling is only a type of cutting without specific relation to coating behavior:

  • Axial depth of cut is high
  • Thermal shock and heating can be key aspects

Response 1: Thank you once again for your help in improving the manuscript. “Trochoidal milling is only a type of cutting without specific relation to coating behavior” – however, it depends on the structure of coating in its large part and type of material to be processed and, as a consequence, on a used tool, modes, and strategy of milling.

The recommended axial depth of cut is recommended to choose in the range of 1-1.5xD (up to 2xD) when D is a mill diameter. Thus, the axial cutting depth ap of 5 mm is lower than the range of recommended values.

The radial depth of cut was revised in the text. With your help, we found the technical mistake that has occurred in the process of manuscript preparation. Now it is of max 2.4 mm that corresponds to the recommendations - 10-20% from the tool diameter.

The thermal shock and heating are the aspects that were taken into account in the development of the cutting mode. As it is known, the release of a large amount of heat during milling of heat-resistant and titanium alloys limits the cutting speed. For cutting titanium alloys, it is recommended using a particular trochoidal strategy to reduce the constant small cutting curve. It should be noted that the trochoidal milling strategy allows cutting with the reduced loads and removing a large volume of the material with a constant width of a chip when the depth of cut ae is variable. The particular geometry of the tool also assists in providing finer chips and constant tool loads in axial and radial directions with an effective number of teeth involved in cutting. Commonly, it is also recommended using the coolant to provide easier chip and heat removal, avoiding secondary involving chips in the process of cutting. However, tests of new materials and/or coatings are always carried out without coolant to exclude the influence of coolant and evaluate the contribution of the coating since the coolant itself increases tool wear resistance, affects contact friction and interaction.

In our experiments, the coated end mills were tested without coolant to provide better observation of wear and titanium adhesion on cutting edges that correlates to the standard tool tests, when the used strategy should allow reducing loads and consequently reducing in thermal shock and heating.

However, the results have shown that wear resistance higher than expected, which correlates to the effective heat removal that can be even improved by using coolant.

The relevant text is added to the manuscript; the position was grounded (in green and marked red).

Point 2: Tool edge must be tough and the main cutting tool brands (Mitsubishi or Emuge Franken) offer all kinds of information about the process. CAM, programming is the main topic, and operation is reduced to slotting. My experience with CFRP in which we used trochoidal was that. Omsrud is making tools using the approaches mentioned.

This aspect in the paper has not relevance, even the title could change.

Response 2: Thank you for your kind comment. We should note that the obtained cutting edges of the end mills were tough enough to provide satisfying wear resistance of the coatings. In our research goals, we did not have an intention in testing cutting tools of various brands, as we find this task more related to the tasks of marketing than to the research wear resistance of coatings in the particular milling conditions. CAM programming is an important part of every complex machining process with a sophisticated trajectory, and we have observed a few works related to the features of CAM-programming. However, these papers (some of them we have mentioned in our reference list, in green and marked red in the list of references) are more related to the formation of professional recommendations for specialists in CAM. They have no focus on testing of the multicomponent coatings in the conditions of real production.

We also have a question on the approach of Onsrud, https://www.onsrud.com/Series/Non-ferrousSlotting-Finishing.asp, we did not find on their website any constructive proposal on coating composition for the carbide tool for slotting titanium alloys. If they have some specific approach, we would like to learn it.

About carbon fiber reinforced polymer, we are glad to hear that you had a great experience in carbon fiber machining. Still, we do not understand how its behavior under the cutting loads with trochoidal milling strategy is correlated with the milling of titanium alloys. The trochoidal milling has a few particularities that make this milling strategy recommended for milling hard-to-machine materials, including heat-resistant and titanium alloys. The main principle is in providing a constant small cutting curve that allows using higher cutting speeds in combination with the significant volumetric performance and, consequently, the whole length of mill working part can be involved in processing. It helps to reduce wear in comparison with the standard strategies of milling. However, the nature of the material destruction in the case of titanium machining is different from carbon fiber milling.

We have as well doubts about not clear suggestions related to the title of our work ''Wear resistance of cathodic-arc deposited multicomponent coatings for carbide end mills at the trochoidal milling of titanium alloy". We have added in the title the words related to the second part of the tests; now it is "Wear resistance and titanium adhesion of cathodic-arc deposited multicomponent coatings for carbide end mills at the trochoidal milling of titanium alloy". We hope that the new title will satisfy the high requirements of the reviewer. The relevant text is added in the introduction and other sections of the paper. (in green and marked red)

Point 3: The long answer is weird. Really tool edge treatment is more important than the coating recipe itself. I have checked the works, all used blasting or OTEC drag grinding. At the States OTEC is not well-known, but similar are in all cutting tool small companies, around Ohio e lot. You did not mention the real challenge, this, and so the paper cannot be accepted in the present form. About 40-505 successful ratios depend on it.  Looking at Platit website, they really make some offers of machines in this line. Some works inside the long argue are better aimed at the rebuttal because they mentioned more aspects of the entire technique.

Response 3: Thank you for your kind evaluation of our modest efforts. The extended answer cannot be weird since we try to ground our position. We would like to point that we have used a tool that has already ground edges as you can see it in the pictures from the scanning electron microscopy (Figure 5, a). This tool has a measured radius of the cutting edge rounding of 6-8 µm before treatment and ~18-22 µm after coating. We would like to notice that it is not understandable why the professional carbide mill with the industrially done geometry and ground cutting edge should be blasted again and re-ground. What is the point in repeating these operations that were already done at industrial conditions? This carbide tool is already a cutting tool. We have coated it in the unit that has pre-treatment operation to clean it from the industrial contaminations before coating.

We know the OTEC equipment that allows dulling the edge of a carbide cutting tool. The purpose of blunting is to stabilize the cutting edge. In the case of the use of brittle materials such as carbide, the sharp edge under the influence of high loads is destroyed and quickly wears out. A simple blunting of the cutting edge to about 15 µm can significantly stabilize the cutting edge, as well as significantly increase the service life and reliability of the cutting tool. However, our tool was already blunted by coating, and if we blunt it mechanically even more before coating, we could have the rounding radius that will not allow us to cut anything. The conducted experiments can confirm our position – the obtained radius of cutting edge was sufficiently blunt to resist the wear during 1150-1580 m of cutting path.

We would point it once again, and it can be seen on the web site of Platit https://www.platit.com/en/products/coating-units/pvd-standard-coating-units/2-pi411/ (411 unit is quite similar to 311 unit) that the used equipment is provided with a module of etching technology (Lateral Glow Discharge) that allows plasma etching with argon and metal ion etching (Ti, Cr). In our case, the coating technology included etching with argon ions with an energy of 500 eV at a pressure of 1 Pa using a non-self-contained gas discharge, which is ignited between the targets. The electron flow between the two targets creates a high-density plasma in which the processed products are immersed. As it is known, high-density plasma effectively cleans or etches products even with complex surfaces. At the same time, a negative potential (voltage bias) of 400 V was supplied to the holder of the cutters to be processed. We would also like to confirm that etching and pre-treatment of the tool plays a key role in the coating deposition. In our case, we used not a billet to produce a tool at University from a to z but an industrially produced end mill already ground in the manufacturing conditions that was properly etched and coated by Platit technology with the developed multicomponent coatings. These coatings resisted to the wear in the conditions of trochoidal milling of titanium during 1100-1580 m of cutting path.

The relevant text is added to the manuscript. (in green and marked red)

Point 4: My opinion is that the paper could be improved much more with some information from manufacturing journals. You can rewrite some parts of the paper, and it can be accepted in manufacturing journals. Technologies and trochoidal milling (trochoidal is a little idea) are not a perfect match.

Summarizing: if the paper would be improved and better discussed the technical aspects using information from results and by others, could be acceptable.

Response 4: Thank you for your kind suggestion. With the help of reviewers, we have provided at least 65 references related to the works of the different research groups and introduced about 70 corrections into the text of manuscript including rewriting of some parts. We do agree that trochoidal milling itself is a complex programming task rather than an object of research in the context of wear resistance of monocoatings with no relation to the material to be processed.  However, our idea was in testing coatings in the conditions close to the most advanced strategies in milling titanium that will give new and more adequate data on wear resistance and titanium adhesion to the cutting edge of the developed coatings in the industrial conditions of trochoidal milling than just a simple cut machining with no idea. We would also point that the structure of the coatings is unique and corresponds to the last achievements. The first coating consists of the adhesive TiN layer and nanolayers of AlTiN; the second is of adhesive TiN layer and nanocomposite of AlTiN/SiN, where nanocrystalline TiAlN grains are in the amorphous SiN matrix; the third proposed coating corresponds to the adhesive CrTiN layer and nanocomposite of AlTiCrN/SiN, where AlTiN and AlCrN nanocrystals are in the amorphous SiN matrix. We have checked once again and did not find any similar work in the international scientific journal databases. There are plenty of works with monolayer coatings but very few with multilayer and nanocomposite structures especially with the proposed content when the complexity of the coating plays a key role in their resistance. The complexity of the coating structure hampers the development of cracks that simply cannot develop through the structure of the coating due to the designed obstacles in the form of multiple layers of coating and nanocrystals in an amorphous matrix of the nanocomposite. In these conditions, an arising crack cannot straightly cross the complex coating chipping the significant part of the coating up to the substrate as it can be seen in the case of the monolayer coatings. In the case of a multi- or nanolayer structure, it stops with the first layer and develops further along the diffusion line between the layers and splits tiny pieces of each layer. In the case of nanocomposite multilayered coatings, the crack hampers by the crystalline grains that change the direction of crack development in amorphous SiN matrix. A more complex nanostructure of the coating makes the development of the crack even more difficult. As it was shown and proven during wear resistance tests, the obtained nanocomposite and multilayered structures of coatings provide changes in the direction and hampers of crack development that has a noticeable impact on wear resistance of the coatings (Figure 1, provided in the file).

The provided experiments in the conditions of titanium alloy trochoidal milling showed that the complexity of the coating structure improves its wear resistance that can be related to the changes in crack development that also become more sophisticated by the complexity of the coating. The provided data correlates with the already published works:

  1. Vereschaka, A.; Grigoriev, S.; Sitnikov, N.; Oganyan, G.; Aksenenko, A.; Batako, A. (December 20th 2017). Delamination and Longitudinal Cracking in Multilayered Composite Nanostructured Coatings and Their Influence on Cutting Tool Wear Mechanism and Tool Life, Novel Nanomaterials - Synthesis and Applications, George Z. Kyzas and Athanasios C. Mitropoulos, IntechOpen, DOI: 10.5772/intechopen.72257.
  2. Vereschaka, A.A.; Grigoriev, S.N.; Sitnikov, N.N.; Batako, A. Delamination and longitudinal cracking in multi-layered composite nano-structured coatings and their influence on cutting tool life. Wear 2017, 390–391, 209–219.
  3. Vereschaka, A.A.; Grigoriev, S.N. Study of cracking mechanisms in multi-layered composite nano-structured coatings. Wear 2017, 378–379, 43-57, https://doi.org/10.1016/j.wear.2017.01.101.

The relevant text with the technical aspects is added to the sections of materials and methods and discussion. (in green and marked red)
